# Challenges and Opportunities in Clinical Diagnostic Routine of Envenomation Using Blood Plasma Proteomics

**DOI:** 10.3390/toxins15030180

**Published:** 2023-02-27

**Authors:** Joeliton dos Santos Cavalcante, Denis Emanuel Garcia de Almeida, Micael Saggion Moraes, Sophia Ribeiro Santos, Pedro Moriel Pincinato, Pedro Marques Riciopo, Laís Lacerda B. de Oliveira, Wuelton Marcelo Monteiro, Rui Seabra Ferreira-Junior

**Affiliations:** 1Graduate Program in Tropical Diseases, Botucatu Medical School (FMB), São Paulo State University (UNESP—Universidade Estadual Paulista), Botucatu 18618-687, São Paulo, Brazil; 2Department of Bioprocess and Biotechnology, School of Agriculture, Agronomic Sciences, São Paulo State University (UNESP—Universidade Estadual Paulista), Botucatu 18610-034, São Paulo, Brazil; 3Graduate Program in Translational Medicine, Drug Research and Development Center (NPDM), Federal University of Ceará (UFC), Fortaleza 60020-181, Ceará, Brazil; 4Graduate Program in Tropical Medicine, Department of Research at Fundação de Medicina Tropical Dr. Heitor Vieira Dourado, Amazonas State University, Manaus 69850-000, Amazonas, Brazil; 5Center for the Study of Venoms and Venomous Animals (CEVAP), São Paulo State University (UNESP—Universidade Estadual Paulista), Botucatu 18619-002, São Paulo, Brazil

**Keywords:** biomarkers, blood plasma, clinical proteomics, human envenoming

## Abstract

Specific and sensitive tools for the diagnosis and monitoring of accidents by venomous animals are urgently needed. Several diagnostic and monitoring assays have been developed; however, they have not yet reached the clinic. This has resulted in late diagnoses, which represents one of the main causes of progression from mild to severe disease. Human blood is a protein-rich biological fluid that is routinely collected in hospital settings for diagnostic purposes, which can translate research progress from the laboratory to the clinic. Although it is a limited view, blood plasma proteins provide information about the clinical picture of envenomation. Proteome disturbances in response to envenomation by venomous animals have been identified, allowing mass spectrometry (MS)-based plasma proteomics to emerge as a tool in a range of clinical diagnostics and disease management that can be applied to cases of venomous animal envenomation. Here, we provide a review of the state of the art on routine laboratory diagnoses of envenomation by snakes, scorpions, bees, and spiders, as well as a review of the diagnostic methods and the challenges encountered. We present the state of the art on clinical proteomics as the standardization of procedures to be performed within and between research laboratories, favoring a more excellent peptide coverage of candidate proteins for biomarkers. Therefore, the selection of a sample type and method of preparation should be very specific and based on the discovery of biomarkers in specific approaches. However, the sample collection protocol (e.g., collection tube type) and the processing procedure of the sample (e.g., clotting temperature, time allowed for clotting, and anticoagulant used) are equally important to eliminate any bias.

## 1. Introduction

Human envenoming by venomous animals represents the main neglected health problem in tropical and subtropical countries [1,2,3,4]. Due to its climatic heterogeneity and different biomes, Brazil harbors a great variety of venomous animals with harmful potential for humans [5,6,7]. The therapeutic procedures for envenoming by venomous animals consist of the administration of antivenoms [8]. An antivenom is a product that contains antibodies, or fragments thereof, that act on the active toxins that are circulating in the victim’s plasma [9]. 

However, the application of an antivenom is not a trivial task due to uncertainties about the identity of the species responsible for the accident [10], the identification of dry bites, the late appearance of symptoms and presence of asymptomatic windows [11], the amount of venom injected, and the dose of antivenom to be used [12]. The lack of knowledge/preparation of the medical team, associated with the lack of diagnostic tools, contributes to these uncertainties as well. Diagnosis, including identifying the animal/genus responsible and the severity of the envenoming, is challenging but crucial in deciding when and how much antivenom to use. Thus far, specific and sensitive diagnostic tools have not yet reached the clinic, resulting in late diagnoses based on clinical manifestations and laboratory tests [1]. This leaves room for errors regarding the administration (or lack thereof) of antivenoms and the amount of antivenom to be administered. The waiting time for decision making based on the observation of signs and symptoms may represent a factor in the progression of the clinical condition from mild to severe illness.

Considering that animal venoms use plasma as a means of biodistribution, their effects on this fluid, as well as the activation of biochemical pathways, modulate the organism’s proteome, providing abundant information and biomarkers for investigation [13]. Serum and/or plasma have attracted substantial attention due to their ease of sample collection and preparation. The dynamics in the abundance of proteins and [14] metabolites, in association with the correlation between disease progression and severity [15,16], can be used for diagnosis, differentiation between pathologies with very similar etiology and clinical manifestations, and monitoring disease progression [17].

Previous studies have addressed the importance and advancements in the field of diagnosis of envenomation by venomous animals, but the use of mass spectrometry, particularly proteomics, has not yet been fully explored [14]. Establishing predictive blood biomarkers for the diagnosis, severity, and complications of envenomations would be highly valuable for prognosis, monitoring disease progression and responses to therapy, and predicting outcomes [14]. Research on plasma biomarkers has increased exponentially in an effort to elucidate the complex pathogenesis of human diseases, leading to the creation of plasma and serum marker panels [13].

However, caution must be taken from the sample collection phase to the interpretation of results in order to fill in gaps and minimize errors and variables during the clinical sample collection and laboratory workbench. The need for new tools in the diagnostic and prognostic field of clinical toxinology and the scarcity of proteomics studies with clinical samples justify the importance of this publication, since it will initiate a discussion on the relevance of the proposed strategy and methodology. Data analysis, including open-source frameworks, software libraries, and tools, will not be considered here, since this is an extensive topic that should be addressed in a future manuscript.

## 2. Diagnostic and Monitoring Tools for Envenomation: An Urgent Need

### 2.1. Snakebite

The diagnosis of snakebite envenomation is hampered by uncertainties regarding the identity of the species that caused the envenomation, and it is often restricted to clinical manifestations and case reports [10,18,19]. The anamnesis of snakebite victims has been based on five basic questions: (i) where were you bitten?; (ii) what time/when were you bitten?; (iii) what were you doing when you were bitten?; (iv) where was the snake and what did it look like?; and, finally, (v) how are you feeling now? [1]. In addition, laboratory tests (such as clotting time) have been used to evaluate hemostasis due to the higher incidence of snake envenomations that induce coagulopathy. These tests include the 20-min whole blood clotting test (20WBCT), bleeding time (TS), prothrombin time (PT), and activated partial thromboplastin time (APTT) [20,21].

Hematological analysis based on cell counts has been used to diagnose snakebite envenomation. Clinically, cases of thrombotic microangiopathy (TMA) with accompanying kidney injury and thrombocytopenia after Bothrops envenomations have been reported in Brazil [22,23,24]. However, markers of microvascular hemolysis and anemia after snake envenomation are not specific yet [25]. The platelet count used to identify thrombocytopenia does not show specificity for the genus and type of venom due to the variability of the action of toxins on platelets, which can cause platelet aggregation or inhibition. Neutrophilia and severe thrombocytopenia have been reported in cases of snakebites that had tissue loss and/or limb amputation [26,27,28], showing a low predictive potential for clinical outcome. This reinforces the need to discover and develop new tools for monitoring snakebites.

Biochemical parameters are used to monitor envenoming, but they have low specificity and sensitivity and may not reflect the actual clinical condition of the patient. For example, creatine kinase (CK) quantification assays are used to identify and monitor myonecrosis caused by venoms. CK has a relatively short half-life, and its activity returns to normal quickly after the cessation of myodegeneration or local necrosis [29]. Although it has been considered a gold standard for assessing muscle damage, evidence has shown that this biomarker does not reflect the amount of tissue damage [29,30]. CK activity depends not only on tissue damage but also on the number of CK molecules present in the plasma/serum and the glutathione concentrations, which tend to decrease during rhabdomyolysis [30].

In addition, asymptomatic cases have been reported in epidemiological studies worldwide [31,32,33,34,35,36]. The absence of clinical manifestations in these cases could be associated with the lack of venom injection, but their diagnosis to decide on the need (or lack thereof) to apply antivenoms and the lack of methods to detect the presence of venom in the patient’s blood may lead to an erroneous interpretation of the case, leading to a misapplication of antivenom that can result in early or late adverse reactions to it [9,11,37].

In this context, Enzyme-Linked Immunosorbent Assay (ELISA) and lateral flow strip assays for the detection of venom in blood samples from patients have been developed [38,39,40]. However, numerous variables come into play with this form of production. One of the problems to be listed regarding the development of these diagnostic kits refers to the cross-reactivity observed between the venoms. Some classes of proteins in each venom overlap, and detection devices are not species/genus-specific and detect a variety of species when using immunological techniques, which may direct the diagnosis to a false-positive or false-negative result [38,39,40]. Another critical factor to be considered is the amount of injected and free venom in the patient’s plasma in relation to the sensitivity of the kit, since the toxins can be highly diluted in the plasma samples and, consequently, outside the detection range [40]. Additionally, there are several toxins that are quickly absorbed into deep tissue, while others act locally and are never absorbed, remaining in the bite/sting site [41].

### 2.2. Scorpion Envenomation 

In scorpion envenomation, the pain caused by the venom of these animals, as well as the morphological characteristics described by patients who take the specimen to the hospital, help in the diagnosis [42]. Although little used in the clinic, enzyme-linked immunosorbent assays of the ELISA type have allowed for the determination of the concentration of venom in patients envenomated by scorpions [43,44,45]. Furthermore, in cases where the victim does not see the animal, the diagnosis can still be made based on the set of elements that involve the occurrence (place of occurrence, pain, clinical signs, and others) [46].

In addition, the use of antivenoms for treatment is based on the clinical condition presented by the victims [2,42], which can lead to errors regarding the administration (or lack thereof) of antivenoms and, in cases where the application is necessary, errors as to the amount of antivenom to be administered. Late systemic manifestations and clinical complications can be detected using electrocardiography to identify cardiac changes [47] and radiography and echocardiography to identify changes in the cardiac area, as well as signs of acute pulmonary edema and other complications [46,48].

### 2.3. Honey Bee Stings

Human envenomation by honey bees can result in a complex pathophysiological picture that includes inflammatory reactions, allergic manifestations, anaphylactic shock, and systemic toxic reactions, depending mainly on the number of stings that the victim received, the age, weight, comorbidities, and medical conditions of the patient [5,49]. However, there is no specific diagnosis or protocol for monitoring the clinical condition of patients. During the multicenter phase I/II clinical trial of the first apilic antivenom for the treatment of Africanized bee stings, several clinical and biochemical parameters were taken into account to monitor treatment and therapeutic success [49,50]. The laboratory findings observed included the presence of stinging at the bite sites, hemodynamic changes, respiratory disorders, elevated levels of CK, C-reactive protein (CRP), fibrinogen, alanine transaminase (ALT), and leukocytosis. In addition, for the first time, plasma samples from patients were analyzed by mass spectrometry, through which it was possible to identify the presence of melittin in bee venom, suggesting that this tool can be used to identify biomarkers for envenomation [50].

### 2.4. Spider Envenomation

The diagnosis of spider envenomation becomes challenging when the spider is not felt or seen by the patient, being completely dependent on clinical and laboratory analysis as well as knowledge of the distribution of the species in the region [51]. The diagnosis based on systemic and/or local symptoms may lead to confusion with other medical conditions that have been or may be diagnosed as recluse bites [52]. Despite being useful as a diagnostic tool in some contexts, this method is primitive and can provide false information, which highlights the need for better diagnoses.

The diagnosis is mainly clinical and focused on the skin wound; however, the clinical team also uses laboratory tests, although nonspecific, to obtain a possible differential diagnosis [52,53,54]. The laboratory diagnosis relies on the presence of several hematological tests (analysis of the red series and WBC) to identify hemolysis and leukocytosis, hemostatic tests (fibrinogen, APTT, PT, and D-dimer assay) to assess the presence of disseminated intravascular coagulopathy, and biochemical tests (ALT, AST, total and direct bilirubin, urea, creatinine, CRP, lactate, lactate dehydrogenase, CK, sodium, potassium, glucose, and venous blood gases) to diagnose and monitor kidney injury [55]. 

Although several spiders cause medically important envenoming, most studies involving the development of tools to identify and quantify toxins are developed for *Loxosceles* sp. and *Latrodectus* sp. Thus, skin exudate samples (passive hemagglutination inhibition test and ELISA) [56,57,58], biopsy and hair samples (competitive ELISA) [56,58,59], and serum [60] have already been used to detect *Loxosceles* venom. Although the possibility of detecting *Loxosceles* venoms has been reported for a long time, their use in the clinic is not yet common knowledge. In addition, assays with serum samples are under development for the diagnosis of envenomation by *Phoneutria*, *Atrax*, and *Hadronyche* spiders. In the case of *Phoneutria* venom, its detection was possible 8 h after experimental envenoming [61], and there are already reports of its detection in human samples [62,63]. The detection of *Atrax* and *Hadronyche* venom in human samples has also been reported [63]. Thus, there is a need for studies that demonstrate the effectiveness of these kits in the laboratory-hospital routine.

## 3. Biomarkers for Envenomation by Venomous Animals: What to Look For?

During the transition from the discovery phase to the verification and validation phases, a different set of quality assessments is required to ensure the analytical validity of the biomarkers. The discovery of new biomarker candidates by MS has evolved, thanks to new applications of MS methods in omics research (lipidomics, metabolomics, and proteomics), allowing for the analysis of complex samples, including blood, urine, and biopsy tissues, and the obtaining of a profile of molecules that indicate a certain pathological scenario [64,65]. Within this context, proteomics has played a distinguished role due to the wide variety of proteins being explored as biomarkers for different human pathogenic conditions [13].

Considering that animal venoms use plasma as a means of biodistribution, their effects on this fluid, as well as the activation of biochemical pathways, modulate the organism’s proteome, providing an abundance of information and biomarkers for investigation [13]. Serum and/or plasma have attracted substantial attention due to their ease of sample collection and preparation, as well as the dynamics in the abundance of proteins and metabolites in association with the correlation between disease progression and severity [15,16,66]. In addition, many of these proteins can be used for diagnosis, differentiation between pathologies with similar etiology and clinical manifestations, and monitoring disease progression [17]

Human envenomation by venomous animals causes changes in plasma protein abundance that are currently monitored by assays that primarily assess changes in hemostasis [20,21]. Additionally, other proteins that are or are not involved in hemostasis have already been identified as potential biomarker candidates but require validation for monitoring clinical complications associated with envenomation [67,68,69,70]. Furthermore, the possibility of identifying circulating animal toxins in plasma samples from patients after multiple bee stings [50] and snake bites [41,71] using proteomics tools has already been reported. Several toxins show a high degree of homology with human blood proteins, which makes accurate identification using MS difficult. Thus, previous experiments can be performed based on the immunodetection of the toxins present in the plasma using Western blot assays [71], and the bands that correspond to the toxins can be easily identified by MS. Therefore, it is also necessary to overcome another challenge with regard to the immunodetection of toxins: the use of specific antibodies that have thus far been obtained from commercial antivenoms [40].

Studies involving mass spectrometry in toxinology have focused on the study of proteins present in animal venoms, converging on (i) the identification of isolated and purified proteins from venoms [72,73,74], (ii) the description of the composition of venoms [75,76,77,78,79,80,81], and (iii) the identification of altered biological processes in tissues after tissue damage caused by snake venom [41,67,68,69,70,82,83]. MS can assist in the development and tracking of target toxins and improve the sensitivity and specificity of ELISA and Western blot assays by identifying toxins circulating in the blood of the victims. This information can be used for the production of antibodies against specific toxins, aiding in the elimination of heterophilic antibodies and reducing the probability of a cross-reaction in the diagnosis. When used in combination, proteomics tools (ELISA, Western blot, and mass spectrometry) have the ability to identify and validate biomarkers for tracking early tissue damage, predicting clinical outcomes, and evaluating therapeutic response (Figure 1). 

The use of specific antibodies in the development of diagnostic tools (ELISA and lateral flow strip assays) for envenoming by venomous animals can accurately determine the type of snakebite incident and classify the severity of the case based on the amount of circulating venom, thus avoiding unjustified administration and unnecessary antivenom [11,40,50,60,61,84]. Consequently, the specificity of the antibodies present in antivenoms is associated with the venoms of species selected for hyperimmunization, and variations in intra/interspecific toxins can result in reduced recognition of toxins from different venoms [85,86,87]. 

Different studies report the use of polyclonal antibodies with low specificity and saturation in high concentrations of snake venom, for example, the polyclonal diagnostic antibodies used to classify the cases of envenomation by hematotoxic and neurotoxic snakes and the polyclonal diagnostic antibodies used for individual species and bees [40,50]. The purification of antibodies from these preparations can compromise the reliability of the number and type of toxins detected, since detection will be dependent on the presence of antibodies that recognize a particular toxin. Therefore, we believe that the study of the production of genus-specific antibodies for venomous animals is indispensable and can be obtained through various techniques of immunization of animals [88,89].

## 4. Biomarker’s Development Phase

Biomarkers are indicator biomolecules that help in early diagnosis, discriminate among different diseases, and provide valuable tools for monitoring the progression/severity of diseases [16,17,90,91,92,93]. The development of biomarkers for human diseases is divided into three phases: discovery, verification, and validation, with the latter being further divided into two stages: analytical validation and clinical validation. As the study progresses through these phases, the number of candidate biomarkers (peptides and proteins) decreases, and they are measured in more samples and subjects [13]. During the transition from the discovery phase to the verification and validation phases, a different set of quality assessments is required to ensure the analytical validity of the biomarkers.

In the discovery phase, a large number of biomarker candidates are identified through an in-depth and untargeted analysis of the proteome of biological samples, which is considered the analyte source that aims to identify as many candidates as possible. However, due to the complexity of obtaining clinical samples, economic cost, and other factors, this phase is carried out with a very small number of patients [13,16,17,90,92,93,94,95]. Peptides, and therefore proteins, are identified by matching experimental tandem mass spectra (MS/MS) to computationally predicted MS/MS spectra.

In the verification phase, assays are performed to validate the abundance of target peptides/proteins in patient samples relative to the control group. One of the most commonly performed tests for this purpose is the addition of synthetic peptides labeled with stable isotopes to the samples. This facilitates the reliable identification and quantification of targeted peptides using mass spectrometry techniques such as selected/multiple reaction monitoring (SRM/MRM) [96]. In the verification phase, the number of samples analyzed depends on the disease, its complexity, previous research, and the analytical testing platform. The number of patients should be based on a power analysis. In addition, the number of subjects ranges from tens to hundreds to confirm the abundance of biomarker candidates [66].

Analytical validity includes the analysis of several parameters such as precision, specificity, sensitivity, recovery, and stability. Precision analysis includes a measure of repeatability, which focuses on investigating within-day variability, and reproducibility, which refers to day-to-day variability [97]. To define the coefficient of variation, repeated analyses under different conditions and at different concentrations are used, and the robustness of a coefficient of variation must be interpreted within the context of what is considered a clinically significant change in the analyte. In addition, verifying that assays yield similar results when performed by different individuals and in different laboratories is part of validating reproducibility [98].

Afterwards, the analytical validation phase is performed to confirm the usefulness of biomarker candidates and their respective assays. To provide a measure of robustness, an expanded cohort of patients used in the earlier stages, and even a cohort of individuals with the same disease/condition not previously analyzed, is considered for analysis. The validation phase also requires a number of patients defined through power analysis, and the number of biomarkers to be tested must also be considered. This can range from tens to thousands of patient samples, which are analyzed by immunological assays such as ELISA, immunohistochemistry, dot blot, Western blot, and others [13,15,66,99,100].

## 5. Preanalytical Variables: Sample Preparation Challenges

Blood is the most commonly used medium in clinical analysis and an important biofluid for researching diagnostic and prognostic biomarkers of human diseases [13,15,16,17,66,90,92,93,95,97,100,101]. An innumerable number of variables can have an immeasurable impact on the results of the analyses (Figure 2). Therefore, they must be minimized by adopting strict criteria and carrying out the collections through the development and implementation of standard operating procedures (SOPs). SOPs must be strictly followed and include detailed criteria and information to be followed from sample collection to processing [102]. This will help to validate and interpret blood-based biomarker results across all studies and will facilitate the implementation of these biomarkers in diagnostic and assay settings. 

The use of blood has its own advantages, such as ease of collection and obtaining, in addition to yield, since more plasma is obtained from an equal amount of whole blood compared to serum. The use of plasma with EDTA or citrate anticoagulant without the addition of protease inhibitors, following the guidelines of the Human Proteome Organization, favors greater reproducibility of the results due to a lower degree of ex vivo degradation [103,104,105,106]. 

Processing time is also an essential issue for sample preparation and experimental design in research [107]. For example, the processing time alters the integrity of human peripheral blood mononuclear cells (PBMC), leading to contamination and activation of granulocytes that can alter the proteome in a biased manner and should be considered during the performance of the studies [108,109]. On the other hand, samples submitted to freeze/thaw cycles do not show degradation, although hemolysis is a known issue associated with delayed sample processing [110]. However, the greatest pre-analytical variation is reported in the centrifugation time and the time from centrifugation to storage, during which tubes are kept at room temperature or cold [111]. Other pre-analytical variables, such as centrifugation conditions, delay time for the first centrifugation, and blood and anticoagulant storage temperatures, contribute significantly to plasma proteomic variation and may result in increased intracellular plasma proteins [107,112].

The removal of major proteins to improve proteomic analysis has been widely studied, and it is a variable of great relevance for the study of biomarkers. Removing major proteins such as albumin allows the detection of other proteins of lower abundance, and as more proteins are removed, additional proteins can be identified [113]. A wide variety of columns for the depletion of major proteins other than albumin (such as IgG, antitrypsin, IgA, transferrin, and haptoglobin) can also be used [114]. However, the investigator must pay attention to the percentage of target-protein capture efficiency, which is related to the manufacturer, workflow, and amount of target proteins from the column selected for depletion. This influences the number of proteins detected and increases the degree of variability in the effectiveness of depletion among individual isoforms of certain proteins [115]. A potential problem with plasma protein depletion is that some non-target proteins may be removed along with the target proteins, either due to their association with the target proteins or because they interact non-specifically with the column [114,116]. Furthermore, the use of depletion columns involves inserting additional sample manipulation steps, leading to losses and introducing more variability into the data [113].

The efficiency, reproducibility, and non-specific binding of different depletion products have been investigated [117,118,119]; however, most studies have focused on removing human serum albumin (HSA) and IgG [119,120,121,122,123]. Different columns for the depletion of these proteins are commercially available, such as those capable of retaining 7 (MARS Hu-7 from Agilent Technologies), 14 (Seppro IgY14 from Sigma Aldrich or the MARS Hu-14 kit from Agilent Technologies), or 20 high-abundant proteins (HAPs) (ProteoPrep20 from Sigma). A topic that requires attention is the ability to immunocapture the largest number of proteins, which should be considered the most efficient depletion system currently available (such as the ProteoMiner approach). Furthermore, the efficiency in obtaining the data must also be considered. For example, the ProteoMiner column induces a reduction in terms of the total number of identified proteins and the total number of peptides, resulting in a high number of proteins (30%) being identified with only one significant peptide [124].

## 6. Sociodemographic Background

Human envenomations by venomous animals are known to have a higher incidence in countries with limited resources, as well as those with lower income and other indicators of poverty [125]. This alerts us to a series of patient and environmental characteristics that are factors that can interfere with proteomic analysis. Factors such as the diversity of socio-epidemiological background, the exposure to various environmental risk factors and infectious agents, ethnicity, lifestyle, diet, alcohol intake, cigarette use, and hormonal variables can significantly alter serum/plasma components [13,126,127].

In patients with neglected diseases, it is also possible to observe a reflection of these sociodemographic factors, since they are endemic diseases in low-income populations. Proteomic studies point to variations in plasma proteins in patients with tuberculosis [128], Chagas disease [129], schistosomiasis [130], and leprosy [131], among others.

Regarding the ethnicity and diet of the groups studied, Brenner et al. (2011) evaluated 54 high-abundance plasma proteins in a multiethnic population of healthy young adults and found significant associations with previously identified dietary patterns. Most of these proteins have functions related to processes such as inflammation and lipid metabolism. Another example is the study by García-Bailo and co-authors [91], who observed that Caucasians had higher mean concentrations of adiponectin than East Asians and South Asians. 

Specifically, in patients with chronic alcohol abuse, an alteration in the microheterogeneity of serum glycoproteins has already been observed, with abnormal isoforms of transferrin and alpha-1 antitrypsin in the serum [132]. In passive smokers, it was possible to identify nine proteins differentially expressed in plasma. Of these, ceruloplasmin and Inter-alpha-trypsin heavy chain H4 inhibitor (ITIH4), which are two acute-phase inflammatory proteins, showed a high number of isoforms and exhibited an increase in their abundance associated with tobacco exposure. This may have been due to a specific proteolytic cleavage or increased instability due to oxidative modifications [133].

Plasma hormonal variables are closely related to sex, as pointed out by the study by Ramsey et al. (2016), which showed that the serum concentrations of 117 of the 171 (68%) molecules studied were associated with sex and/or female hormonal status. The study took into consideration variables such as age, Body Mass Index, medication use, lifestyle, health, and other relevant demographic variables. The biomarker studies produced up to 40% of false leads when the patient and control groups were not matched for sex, and up to 41% of false results when premenopausal women were not matched for oral contraceptive pill use. Failure to account for sex and female hormonal status as important sources of variability in serum molecule concentrations can result in confounding these variables with disease status, reducing the power to detect differences, and contributing to poor analysis performance [126].

Consequently, the selection of research subjects should follow comparable sex ratios, age groups, demographic characteristics, and dietary considerations. The choice and careful recording of these factors are essential to minimize pre-analytical variation, since the lack of detailed information can lead to misinterpretations of the results obtained. Therefore, meticulous monitoring of all variables is extremely important for data reproducibility and for correct analysis/interpretation [132].

## 7. Clinicopathological Background

Clinicopathological details, mainly information about the time between envenomation and arrival at the hospital, local and systemic manifestations, clinical complications, the amount of antivenom administered, and use of other drugs, are crucial during the analysis of the plasma proteome of patients. The lack of detailed information can lead to erroneous interpretations of the results obtained. Therefore, meticulous screening of all clinical variables is extremely important to obtain reproducible results and for further analysis/interpretation of the results. To this end, clinical evaluation forms must be completed as clearly and completely as possible and accompany the samples to be entered in the clinical study. Obtaining well-annotated samples is very challenging, but assembling stable biobanks/biorepositories from collected and stored samples from a large number of samples with comprehensive clinical-pathological and socio-epidemiological information can speed up the studies [94,134].

## 8. Other Directions for Studies of Biomarkers in Envenomations

In recent years, there has been a tremendous advance in the use of clinical proteomics in the search for new biomarkers. Critically, the development and refinement of techniques that allow identifying and validating the use of biomarkers has paved the way for preliminary studies that investigate their usefulness in clinical settings [13,16,17,90,93]. However, in clinical toxinology, the process of searching for new molecules for use in diagnosis, monitoring, and prognosis is still in its early stages. Several significant obstacles begin to emerge from the design of studies for this purpose, but the efforts will bring immeasurable gains in terms of the implementation and usefulness of these molecules in clinical care for envenomation by venomous animals.

The number of species within the same genus can be a critical challenge for studies of biomarkers in envenomation by venomous animals. As observed in experimental studies, different biomarker candidates may be associated with the species that caused the envenomation, requiring the search for candidates that overlap at the genus level [67,68,69]. There is an opportunity for high specificity if the biomarker target is not endogenous, for example, specific toxins with markers [41,50,71]. This highlights the need for large-scale studies and clinical trials involving the recruitment of participants in different regions affected by different species and genera of venomous animals for discovery, validation, and standardization purposes.

Furthermore, the ability to actually target and separate the biomarkers associated with clinical complications from those that are not could be a potential challenge. One of the limitations encountered so far in studies is that they display a map of the proteome differentially expressed at a single moment during disease progression compared to healthy individuals or individuals who did not present a specific event [37,67,68,69,70]. Single-point analysis (case versus control) in the case of envenomations by venomous animals, which involves obtaining an overview of biomarkers at the moment the patient enters the hospital service, will help to identify possible diagnostic markers.

However, longitudinal studies involving repeated observations of the same individuals over different time periods, such as different intervals after antivenom treatment until hospital discharge, may provide valuable additional information about changes in biomarkers and their usefulness as monitoring or prognostic markers [135,136]. In addition, the longitudinal analysis of molecules associated with clinical complications can be identified in this way, with the change in the abundance of these biomarkers being monitored and validated as a potential predictor [83].

While blood-based proteomic analyses have been the most common avenue of investigation, other samples such as exudate and the contents present in blisters offer a unique opportunity to further refine biomarkers and increase their sensitivity, specificity, and overall reliability [41,137]. Indeed, studies comparing the efficacy of plasma and other fluid-based biomarkers (or their combinations) will be of great interest.

## 9. Conclusions

Human envenomation by venomous animals is of great epidemiological importance in the world due to the high annual number of cases and deaths, which is aggravated by the lack of more precise and specific diagnostic tools for monitoring associated complications. The development of simple, accurate, low-cost, and stable diagnostic, monitoring, and predictive tools for clinical complications and their implementation in developing countries are crucial due to the high incidence of complications and deaths resulting from envenomation by venomous animals, mainly due to the lack of early and predictive diagnosis and timely treatments. Therefore, without a doubt, the use of proteomics in the clinic will be one of the most promising applications for the identification of diagnostic and prognostic biomarkers and the development of predictive tools for complications and envenomation outcomes, which can effectively improve diagnosis and therapy.

## Figures and Tables

**Figure 1 toxins-15-00180-f001:**
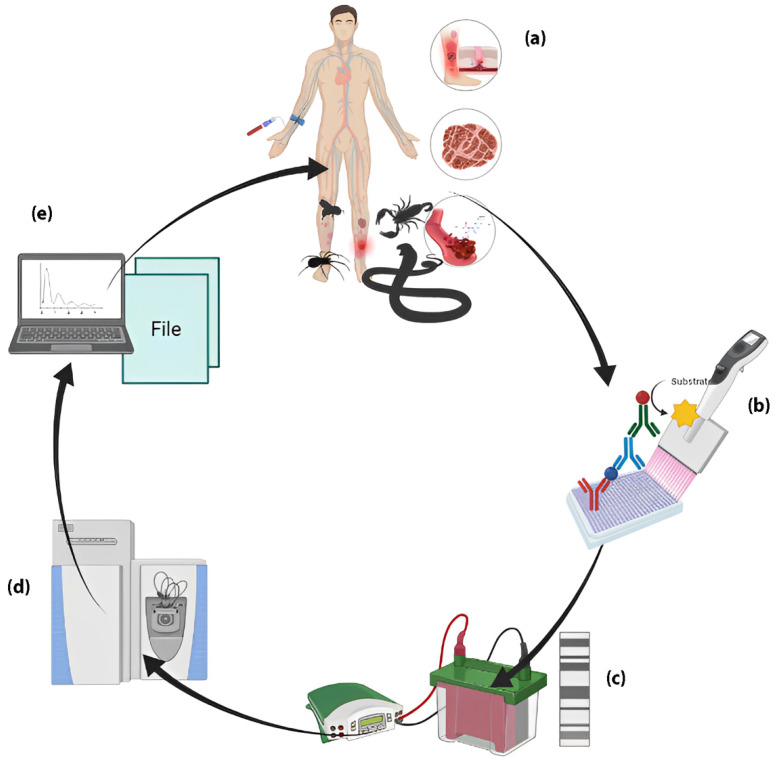
Suggested workflow for discovery of biomarkers (toxins and toxin fragments) in victims of venomous animals. (**a**,**b**) After collecting plasma samples from patients, the concentration of venom in the plasma should be explored using ELISA assays. (**c**) Then, plasma proteins must be fractionated using SDS-PAGE and analyzed using Western blot with antibodies specific to the target venoms. This will provide the catalog of toxins and their fragments present in the patients’ plasma. (**d**) Bands and spots corresponding to toxins and their fragments identified using Western blot must be excised, digested, and analyzed using LC–MS/MS. (**e**) Peptides should be identified and quantified using bioinformatics tools. Created with BioRender.com (accessed on 8 September 2022) by Joeliton S. Cavalcante.

**Figure 2 toxins-15-00180-f002:**
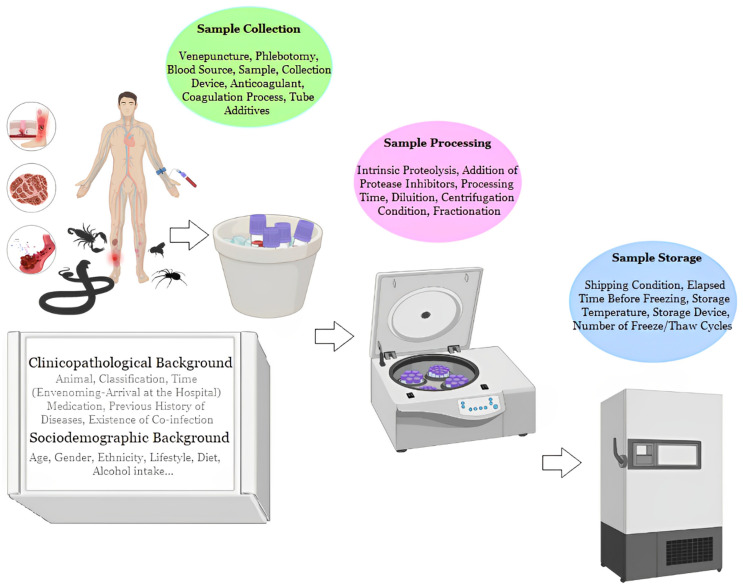
Different sources of pre-analytical variability for proteomics biomarker discovery. Created with BioRender.com (accessed on 8 September 2022) by Joeliton S. Cavalcante.

## Data Availability

Not applicable.

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
