# Peer review of "Challenges and Opportunities in Clinical Diagnostic Routine of Envenomation Using Blood Plasma Proteomics"

_toxins, 2023, doi:10.3390/toxins15030180_

Round 1
Reviewer 1 Report
The manuscript entitled: “Biomarkers from Envenomation by Venomous Animals: Challenges and Opportunities in Clinical Diagnostic Routine Using Blood Plasma Proteomics” attempts to provide a state-of-the-art of the use of protein biomarkers for diagnosis and treatment of animal envenomation. The subject is relevant and the authors mention several important points that can bring attention to diagnostic issues in clinical toxicology. Unfortunately, it is not an extensive revision and therefore I believe it would be more suited for an opinion/discussion manuscript than a review. The organization of the writing is somewhat complicated, which makes it hard to identify the relevant points of the manuscript.
Major comments:
The tittle is a bit misleading because it suggests that the manuscript will provide with a list of the currently used or available biomarkers, and this is not the case. I suggest deleting the first part, for it to read something like: “Challenges and Opportunities in Clinical Diagnostic Routine of envenomation Using Blood Plasma Proteomics”
Line 106. In this context, it is very important to mention the possible late appearance of symptoms and presence of asymptomatic windows that can last several hours. This can happen in several cases of elapid envenomation such as coral snakes, among many others. If this is not mentioned, it can increase the mistakes in diagnosis that the authors are attempting to bring attention to.
Line 111. Using the term adverse reactions is not clear, because it can refer to antivenom or venom adverse reactions, please clarify.
Line 118. It is relevant to mention, even if it is brief, venom and antivenom pharmacokinetics. Several toxins that are quickly absorbed into deep tissue, and others that act locally and are never absorbed and remain in the bite/sting site. This can dramatically affect the possibility of detecting venom molecules in plasma.
Line 122 (Section 2.2). All of the statements in this section can also be applied to snake envenomation. How is scorpion envenomation different to snake envenomation? why separate them?
Line 153 (Section 2.4). Most of the section is focused on Loxosceles bites, but that is not clearly stated until the last paragraph. I suggest clarifying this at the beginning or adding information regarding other relevant spider species such as Latrodectus, which is a very different case.
Line 185. The authors list the phases and then start describing the final stage (clinical validation) before all the others. This is extremely confusing. I suggest describing the phases in the same other they were initially listed.
Line 219. The authors have been discussing biomarkers on section 3, but they don’t define a biomarker until the beginning of section 4. I suggest inverting the order of sections three and four. This will allow a more logical flow for the reading.
Line 250. I don't understand what is this challenge, until when? Please rewrite for clarity.
Figure 1. I believe this figure should be better explained in the caption. Maybe add letters or numbers per section.
Line 275. The authors mention that there are antibody-based tools that can identify the species that caused the envenomation and predict severity. Unfortunately they do not mention or cite any of this tools. It would be relevant to cite if any of these are already in use either in clinical practice or in research.
Line 277. The authors claim that the amount of circulating venom can be used as a predictor of severity for snakebites. Please provide evidence that this is the case.
Line 289. This statement seems to suggest that there is a need for development of a diagnostic antibody for each possible venomous species. Please clarify if this is the case and if so, if you believe this to be feasible or there would be other alternatives.
Line 367. This last sentence seems to be unrelated to the rest of the section.
Line 369. It is not clear what is the implication of this finding for the diagnosis of envenomation.
Line 116. This sentence is not clear. Is the mentioned cross-reactivity between venoms an advantage or a disadvantage? If it can be both please clarify in what sense.
Finally, it is imperative to include an English language revision. The use of very long lines, particularly in section 3, makes the reading very challenging.
Examples:
-Line 53: The sentence starts with “Due to..” but does not conclude, so it is incomplete. Please rewrite to clarify.
-Line 101: The sentence starting with “Although it has been considered a gold standard for assessing muscle damage…” is extremely long and hard to understand.
-Line 201: The sentence “In the verification phase, assays that seek to validate the abundance of target peptides/proteins in patient samples in relation to the control group.” is missing the verb.
-Lines 206 and 215: These sentences are way too long. They could easily be divided in two or even three each, for clarity.
Some minor comments:
Line 80: Change the word “stung” to “bitten”
Line 86: Term TMA has not been defined before.
Line 170: Please change “…and quantify toxins in Loxosceles sp.” To something like “and quantify toxins are developed for Loxosceles sp.”
Line 215. What is meant by “power analysis”?
Line 235. I suggest changing “Due to” for “Also due to”
Line 241: The words “by assays” are repeated.
Line 275. Please substitute the word “poisoning” for “envenoming”.
There are a few terms that are not defined throughout the text, Such as HSA (Line 338), HAPs (Line 341) and TMA (Line 86). Even if they are commonly used in the field, it would be good to have them defined when they are first mentioned in the text.
Line 366. I suggest using a more specific term instead of “whites”, such as “Caucasians”.
Author Response
Reviewer 1
The manuscript entitled: “Biomarkers from Envenomation by Venomous Animals: Challenges and Opportunities in Clinical Diagnostic Routine Using Blood Plasma Proteomics” attempts to provide a state-of-the-art of the use of protein biomarkers for diagnosis and treatment of animal envenomation. The subject is relevant and the authors mention several important points that can bring attention to diagnostic issues in clinical toxicology. Unfortunately, it is not an extensive revision and therefore I believe it would be more suited for an opinion/discussion manuscript than a review. The organization of the writing is somewhat complicated, which makes it hard to identify the relevant points of the manuscript.
Response: All points have been clarified and manuscript was reviewed by a professional in scientific reviews in the medical and biological field.
Major comments:
The tittle is a bit misleading because it suggests that the manuscript will provide with a list of the currently used or available biomarkers, and this is not the case. I suggest deleting the first part, for it to read something like: “Challenges and Opportunities in Clinical Diagnostic Routine of envenomation Using Blood Plasma Proteomics”
Response: Thank you. We agree with you. The title has been changed to “Challenges and Opportunities in Envenomation Clinical Diagnostic Routine Using Blood Plasma Proteomics”
Line 106. In this context, it is very important to mention the possible late appearance of symptoms and presence of asymptomatic windows that can last several hours. This can happen in several cases of elapid envenomation such as coral snakes, among many others. If this is not mentioned, it can increase the mistakes in diagnosis that the authors are attempting to bring attention to.
Response: The sentence was rewritten and this information included.
Line 111. Using the term adverse reactions is not clear, because it can refer to antivenom or venom adverse reactions, please clarify.
Response: The sentence has been changed and the information was added.
Line 118. It is relevant to mention, even if it is brief, venom and antivenom pharmacokinetics. Several toxins that are quickly absorbed into deep tissue, and others that act locally and are never absorbed and remain in the bite/sting site. This can dramatically affect the possibility of detecting venom molecules in plasma.
Response: Thanks for the suggestion. This information has been added.
Line 122 (Section 2.2). All of the statements in this section can also be applied to snake envenomation. How is scorpion envenomation different to snake envenomation? why separate them?
Response: Thanks for the concern and point of view. However, we believe that because they are completely different animals, and both are important for public health, each one deserves its due space in our work, in order not to confuse the reader who does not know one of the species well.
Line 153 (Section 2.4). Most of the section is focused on Loxosceles bites, but that is not clearly stated until the last paragraph. I suggest clarifying this at the beginning or adding information regarding other relevant spider species such as Latrodectus, which is a very different case.
Response: Thanks for the suggestion. We have also inserted mentions for other spider genera relevant to human health.
Line 185. The authors list the phases and then start describing the final stage (clinical validation) before all the others. This is extremely confusing. I suggest describing the phases in the same other they were initially listed.
Response: The sentence was removed and inserted in the descriptive part about analytical validation.
Line 219. The authors have been discussing biomarkers on section 3, but they don’t define a biomarker until the beginning of section 4. I suggest inverting the order of sections three and four. This will allow a more logical flow for the reading.
Response: The order of sections 3 and 4 were reversed as suggested.
Line 250. I don't understand what is this challenge, until when? Please rewrite for clarity.
Response: Perfect. I was really confused. The sentence has been rewritten
Figure 1. I believe this figure should be better explained in the caption. Maybe add letters or numbers per section.
Response: The figure caption has been changed as per suggestion.
Line 275. The authors mention that there are antibody-based tools that can identify the species that caused the envenomation and predict severity. Unfortunately, they do not mention or cite any of this tool. It would be relevant to cite if any of these are already in use either in clinical practice or in research.
Response: Articles that evidence this fact were referenced.
Line 277. The authors claim that the amount of circulating venom can be used as a predictor of severity for snakebites. Please provide evidence that this is the case.
Response: Articles that evidence this fact were referenced.
Line 289. This statement seems to suggest that there is a need for development of a diagnostic antibody for each possible venomous species. Please clarify if this is the case and if so, if you believe this to be feasible or there would be other alternatives.
Response: This sentence was rewritten.
Line 367. This last sentence seems to be unrelated to the rest of the section.
Response: This sentence was removed.
Line 369. It is not clear what is the implication of this finding for the diagnosis of envenomation.
Response: This alerts to a set of characteristics of the patient and the environment, which are factors that can interfere with proteomic analysis. This fact is highlighted in the introductory paragraph of the section.
Line 116. This sentence is not clear. Is the mentioned cross-reactivity between venoms an advantage or a disadvantage? If it can be both please clarify in what sense.
Response: The sentence was rewritten to clarify this point.
Finally, it is imperative to include an English language revision. The use of very long lines, particularly in section 3, makes the reading very challenging.
Examples:
-Line 53: The sentence starts with “Due to..” but does not conclude, so it is incomplete. Please rewrite to clarify.
-Line 101: The sentence starting with “Although it has been considered a gold standard for assessing muscle damage…” is extremely long and hard to understand.
-Line 201: The sentence “In the verification phase, assays that seek to validate the abundance of target peptides/proteins in patient samples in relation to the control group.” is missing the verb.
-Lines 206 and 215: These sentences are way too long. They could easily be divided in two or even three each, for clarity.
Response: Thank you. The manuscript was again revised by an experienced native English proofreader.
Some minor comments:
Line 80: Change the word “stung” to “bitten”
Response: The suggestion was accepted and the term “stung” was replaced by “bitten”.
Line 86: Term TMA has not been defined before.
Response: The term has been defined
Line 170: Please change “…and quantify toxins in Loxosceles sp.” To something like “and quantify toxins are developed for Loxosceles sp.”
Response: The suggestion was accepted.
Line 215. What is meant by “power analysis”?
Response: Power can be defined as the probability of finding a real difference, if it exists. 80% or 0.8 is considered an acceptable value for power. The power analysis is performed independently for each protein. Given the sample size, the variance of abundance values and the size of the difference we want to detect, we can calculate the power. Also, for a given power of 80% we can determine how many samples are required to ensure we find a difference if it actually exists.
Line 235. I suggest changing “Due to” for “Also due to”
Response: The suggestion was accepted and the term “Due to” was replaced by “Also due to”.
Line 241: The words “by assays” are repeated.
Response: The repeat has been removed
Line 275. Please substitute the word “poisoning” for “envenoming”.
Response: The suggestion was accepted and the term “poisoning” was replaced by “envenoming”.
There are a few terms that are not defined throughout the text, Such as HSA (Line 338), HAPs (Line 341) and TMA (Line 86). Even if they are commonly used in the field, it would be good to have them defined when they are first mentioned in the text.
Response: The terms have been defined.
Line 366. I suggest using a more specific term instead of “whites”, such as “Caucasians”.
Response: The suggestion was accepted and the term “whites” was replaced by “Caucasians”.

Reviewer 2 Report
Human envenomation by venomous animals is of great epidemiological importance in the world. Due to the lack of high accuracy, high sensitivity, simple and convenient detection technicals at the present stage, the identification of venomous animal infringement is greatly limited. Here, the manuscript review the state of on routine laboratory diagnoses of envenomation by snakes, scorpions, bees, and spiders, and review of diagnostic methods and the challenges encountered. Overall, this is a fairly complete overview, but there are some details that need to be improved.
1, some venomous animals have special toxins, such as snake venom alpha neurotoxin, scorpion alpha long chain toxin, etc., which might be important biomarkers. This should be added.
2, Since they are biomarkers from venomous animals, should a detailed description of common venomous animals and their venom be provided? Such as the centipede and its venom.
3, The authors should introduce the existing animal invasion detection technology and their advantages and disadvantages.
4, The full text of the writing needs to be improved, such as 350-352 lines and 394-396 lines should be repeated, which is intolerable
Author Response
Reviewer 2
Human envenomation by venomous animals is of great epidemiological importance in the world. Due to the lack of high accuracy, high sensitivity, simple and convenient detection technicals at the present stage, the identification of venomous animal infringement is greatly limited. Here, the manuscript review the state of on routine laboratory diagnoses of envenomation by snakes, scorpions, bees, and spiders, and review of diagnostic methods and the challenges encountered. Overall, this is a fairly complete overview, but there are some details that need to be improved.
Response: Thank you for your considerations. One of the major concerns of our research group is the misuse of antivenoms, which often occurs due to the wrong diagnosis of the animal. We are working hard to seek alternatives for rapid diagnosis to help minimize these cases.
1, Some venomous animals have special toxins, such as snake venom alpha neurotoxin, scorpion alpha long chain toxin, etc., which might be important biomarkers. This should be added.
Response: Indeed, there are already many studies aimed at identifying biomarkers in animal venoms. You mentioned some like snake venom alpha neurotoxin, scorpion long chain alpha toxin. However, the main focus of the work is not to detail these biomarkers, but to the reader about the importance of deepening research in the area.
2, Since they are biomarkers from venomous animals, should a detailed description of common venomous animals and their venom be provided? Such as the centipede and its venom.
Response: There are many venom-producing animals in nature. We focused this work on those that really present a public health problem to man, as they cause thousands of deaths annually around the world.
3, The authors should introduce the existing animal invasion detection technology and their advantages and disadvantages.
Response: Thanks for the suggestion but this was not the focus of the work
4, The full text of the writing needs to be improved, such as 350-352 lines and 394-396 lines should be repeated, which is intolerable
Response: Thank you very much. The sentences have been modified and rewritten to avoid any repetition.

Round 2
Reviewer 1 Report
All of my previous comments and concerns have been adequately addressed. I thank the authous for their thorough revision of the manuscript and hope my comments contributed to its improvement.
Reviewer 2 Report
The revised version basically meets the publishing requirements of Toxins.